# Automated system for diagnosing endometrial cancer by adopting deep-learning technology in hysteroscopy

Yu Takahashi[1], Kenbun Sone[1]*, Katsuhiko Noda[2], Kaname Yoshida[2],
Yusuke Toyohara[1], Kosuke Kato[1], Futaba Inoue[1], Asako Kukita[1], Ayumi Taguchi[1],
Haruka Nishida[1], Yuichiro Miyamoto[1], Michihiro Tanikawa[1], Tetsushi Tsuruga[1],
Takayuki Iriyama[1], Kazunori Nagasaka[3], Yoko Matsumoto[1], Yasushi Hirota[1],
Osamu Hiraike-Wada[1], Katsutoshi Oda[4], Masanori Maruyama[5], Yutaka Osuga[1],
Tomoyuki Fujii[1]

1 Department of Obstetrics and Gynecology, Graduate School of Medicine, The University of Tokyo, Tokyo,
Japan, 2 Predicthy LLC, Kanagawa, Japan, 3 Department of Obstetrics and Gynecology, Teikyo University
School of Medicine, Tokyo, Japan, 4 Division of Integrative Genomics, Graduate School of Medicine, The
University of Tokyo, Tokyo, Japan, 5 Maruyama Memorial General Hospital, Saitama, Japan

* ksone5274@gmail.com

doi.org/10.1371/journal.pone.0248526

Madrid, SPAIN

**Data Availability Statement:** All relevant data are
within the paper and its Supporting Information
files.

## Abstract

Endometrial cancer is a ubiquitous gynecological disease with increasing global incidence.
Therefore, despite the lack of an established screening technique to date, early diagnosis of
endometrial cancer assumes critical importance. This paper presents an artificial-intelli-
gence-based system to detect the regions affected by endometrial cancer automatically
from hysteroscopic images. In this study, 177 patients (60 with normal endometrium, 21
with uterine myoma, 60 with endometrial polyp, 15 with atypical endometrial hyperplasia,
and 21 with endometrial cancer) with a history of hysteroscopy were recruited. Machine-
learning techniques based on three popular deep neural network models were employed,
and a continuity-analysis method was developed to enhance the accuracy of cancer diagno-
sis. Finally, we investigated if the accuracy could be improved by combining all the trained
models. The results reveal that the diagnosis accuracy was approximately 80% (78.91–
80.93%) when using the standard method, and it increased to 89% (83.94–89.13%) and
exceeded 90% (i.e., 90.29%) when employing the proposed continuity analysis and combin-
ing the three neural networks, respectively. The corresponding sensitivity and specificity
equaled 91.66% and 89.36%, respectively. These findings demonstrate the proposed
method to be sufficient to facilitate timely diagnosis of endometrial cancer in the near future.

## Introduction

Endometrial cancer is the most common gynecologic malignancy, and its incidence has
increased significantly in recent years [1]. Patients demonstrating early symptoms of the dis-
ease or suffering from low-risk endometrial cancer can be prescribed a favorable prognosis.

**Funding:** This work was financially supported by Japanese Foundation for Research and Promotion of Endoscopy. The funders had no role in study design, data collection and analysis, decision to publish, or preparation of the manuscript.

**Competing interests:** Kenbun Sone has a joint research agreement with Predicthy LLC. Katsuhiko Noda and Kaname Yoshida are members of Predicthy LLC. The other authors have no competing interests to disclose. This does not alter our adherence to PLOS ONE policies on sharing data and materials.

However, patients diagnosed with endometrial cancer in its later stages have very few treatment or prognosis options available at their disposal [2]. Additionally, patients demonstrating conditions, such as atypical endometrial hyperplasia (AEH), precancerous condition of endometrial cancer, or stage 1A endometrial cancer without muscle invasion, are eligible for progestin therapy. Accordingly, they might potentially be able to preserve their fertility [3]. Therefore, early diagnosis of endometrial cancer assumes paramount importance. Cervical cytology through pap smear is a common screening method employed in cervical cancer diagnosis [4]. However, endometrial cytology is not a reliable screening technique because its underlying procedure comprises a blind test, results of which may lead to a large number of false negatives. Although the standard diagnostic procedure for endometrial cancer involves endometrial biopsy performed via dilation and curettage, a clinically established screening for endometrial cancer does not exist to date [5]. Hysteroscopy is, in general, considered the standard procedure for examining endometrial lesions by directly evaluating the uterine cavity. It is noteworthy that recent studies have suggested that hysteroscopy can be considered an effective technique for accurate endometrial-cancer diagnosis [6,7]. We have previously reported the usefulness of biopsy through office hysteroscopy with regard to endometrial cancer [8].

Artificial intelligence (AI) enables computers to perform intellectual actions, such as language understanding, reasoning, and problem solving, on behalf of humans. Machine learning is a cutting-edge approach for developing AI models based on the scientific study of algorithms and statistical models used by computer systems to perform tasks efficiently. The use of an appropriate AI model also enables computers to learn patterns in available datasets and make inferences from given data without the need for providing explicit instructions [9]. The deep neural network (DNN) facilitates realization of deep-learning concepts. Additionally, it is a machine-learning method that focuses on the use of multiple layers of neural networks [10–12]. From the machine-learning perspective, a neural network comprises a network or circuit of artificial neurons or nodes [13]. Deep learning has garnered much interest in the medical field because deep-learning techniques are particularly suitable for image analysis. They are used for classification, image quality improvement, and segmentation of medical images. Conversely, shallow machine learning is not suitable for image recognition [14]. Recently, several systems developed for use in medical applications, such as image-based diagnosis and radiographic imaging of breast and lung cancers [15,16], have adopted AI models based on the implementation of DNN technology. Numerous examples of such systems employing endoscopic images in the diagnosis of gastric and colon cancer have been reported. However, no such system has been developed with specific focus on endometrial cancer [17,18]. In general, a voluminous amount of data is required for training a model to be highly accurate; this can be possible only if a large number of participants is considered. With the development of deep learning, it is expected that the accuracy rate will be high when the number of samples is large. However, when deep learning is applied to the medical field, some diseases must be analyzed with a small number of samples. Therefore, the challenge for medical AI research is to develop a system analysis method to improve accuracy with a small number of samples.

Therefore, the proposed study aims at developing a DNN-based automated endometrial-cancer diagnosis system that can be applied to hysteroscopy. Hysteroscopy has not yet found widespread utilization in diagnostic applications for endometrial cancers. This further limits the availability of training data for DNNs. Thus, the objective of this study is to develop a method that facilitates high-accuracy endometrial-cancer diagnosis, despite the limited number of cases available in the training dataset. In addition, the purpose of this research is to establish a system for shifting to large-scale research in the future. Because no standard method has been established for use in such scenarios to date, this study focuses on the determination of an optimum method.

In this study, we have achieved a high accuracy for diagnosis of endometrial cancer by hysteroscopy with such a small sample in using deep learning.

## Materials and methods

### Dataset overview

The data utilized in this study were extracted from videos of the uterine lumen captured using a hysteroscope. The breakdown of the extracted data is presented in Table 1 and Fig 1. The shortest video lasted 10.5 s, whereas the longest lasted 395.3 s. The corresponding mean and median durations equaled 77.5 s and 63.5 s, respectively. Because the videos were captured using different hysteroscopic systems with no consistency in terms of the resolution and image position, only parts of the captured images were extracted with the resolution reduced to 256 × 256 px for Xception [19] and 224 × 224 px for MobileNetV2 [20] and EfficientNetB0 [21]. Representative hysteroscopic images pertaining to each condition are depicted in Fig 1. The said hysteroscopic data were collected from 177 patients recruited in this study. These patients had a history of hysteroscopy, and they were categorized into five groups—those demonstrating conditions of a normal endometrium (60), uterine myoma (21), endometrial polyp (60), AEH (15), and endometrial cancer (21) (S1 Table). The above-mentioned data collection was performed at the University of Tokyo Hospital between 2011 and 2019 after obtaining prior patient consent and approval from the Research Ethics Committee at the University of Tokyo (approval no. 3084-(3) and 2019127NI-(1)).

The consent was obtained by allowing the patients to opt-out. Patients were identified as those showing symptoms such as abnormal bleeding or menorrhagia, which required them to visit the outpatient department for the diagnosis of intrauterine lesion via hysteroscopy. The pathological diagnosis of AEH and endometrial cancer was obtained by biopsy or surgery. Normal endometrium, hysteromyoma, and endometrial polyp were diagnosed based on endometrial cytology, histology, hysteroscopic findings by a gynecologist, imaging findings such as MRI and ultrasound findings, and clinical course.

### Training and evaluation data

The prepared videos were divided into four groups at random—three groups were used for training, and the remaining group was used for evaluation. The four groups were denoted pair-A, pair-B, pair-C, and pair-D and used for cross validation. S2 Table presents the number of training and evaluation videos for each pair. The accuracy of the trained model was evaluated based on image and video units. Owing to the limited number of cases available for this study, we defined two classes—"Malignant" and "Others"—for training and prediction. The "Malignant' class included AEH and cancer, whereas the "Others" class included uterine

**Table 1. Images extracted from hysteroscopy videos per each disease category.**

|  | Still image | Video image |
|---|---|---|
| Total number | 411, 800 images | 177 videos |
| Clinical diagnosis n (%) |  |  |
| Normal | 113,357 (27.5%) | 60 (33.8.%) |
| Polyp | 143,449 (34.8%) | 60 (33.8%) |
| Myoma | 45,037 (11.0%) | 21 (11.8%) |
| Atypical endometrial hyperplasia | 42,146 (10.2%) | 15 (8.4%) |
| Endometrial cancer | 67,811 (16.4%) | 21 (11.8%) |

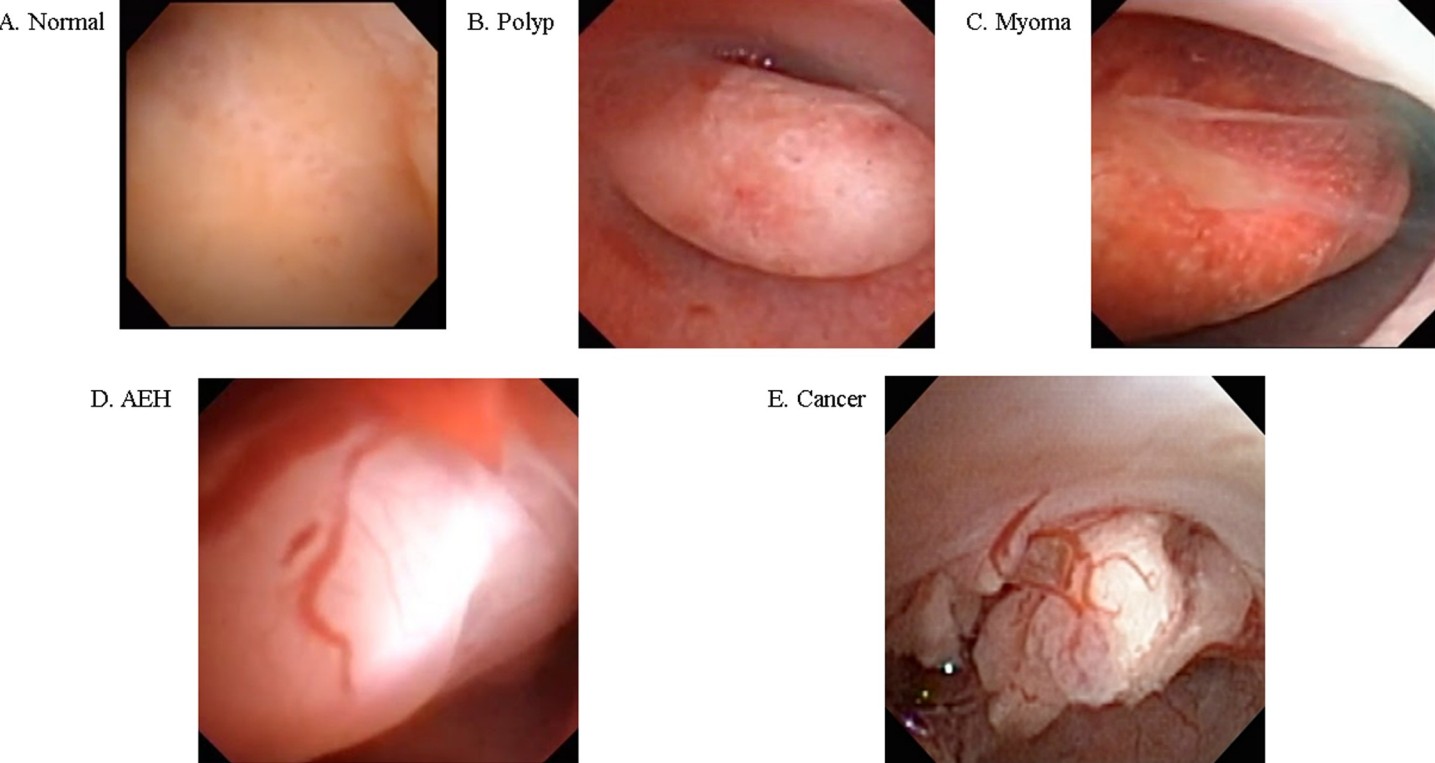

**Fig 1.** Representative images of detected lesions for conditions of (A) normal endometrium; (B) endometrial polyp; (C) myoma; (D) AEH, and (E) endometrial cancer.

myoma, endometrial polyps, and normal endometrium. As listed in S3 Table, the "Malignant" class comprised 36 videos and 109,957 images, whereas the "Others" class comprised 141 videos and 301,843 images. The overall architecture of the model developed in this project is depicted in Fig 2.

**Training data.** The training data pertaining to the malignant class were distributed into the following two sets (Fig 3A).

Set X: comprising all frames included in the video stream.

Set Y: comprising images excluding the outside of the uterine cavity, such as the cervical and extrauterine images from Set X.

The number of frames within each set is listed in S3 Table.

**Evaluation methods.** In this study, the accuracy of the trained model was evaluated in two ways—image-by-image evaluation and video-unit evaluation. During image-by-image evaluation, 100 images that clearly included the lesion site were extracted from the hysteroscopic video of each patient diagnosed with a malignant tumor (Fig 3B). For patients diagnosed with benign and normal tumors, all frames were used during evaluation. In contrast, during video-unit evaluation, the judgment was made depending on the number of consecutive frames classified as "Malignant" in a given video stream (Fig 3C) (Continuity analysis). The threshold value of 50 was set for the number of consecutive frames in accordance with the results of a pre-study we performed, as described in Fig 4A. The threshold was taken from the points where the malignant score intersects with the other scores rather than the point where the average of two scores was the best, because the threshold should be set lower to reduce oversight cases in the actual clinical devices (Fig 4A).

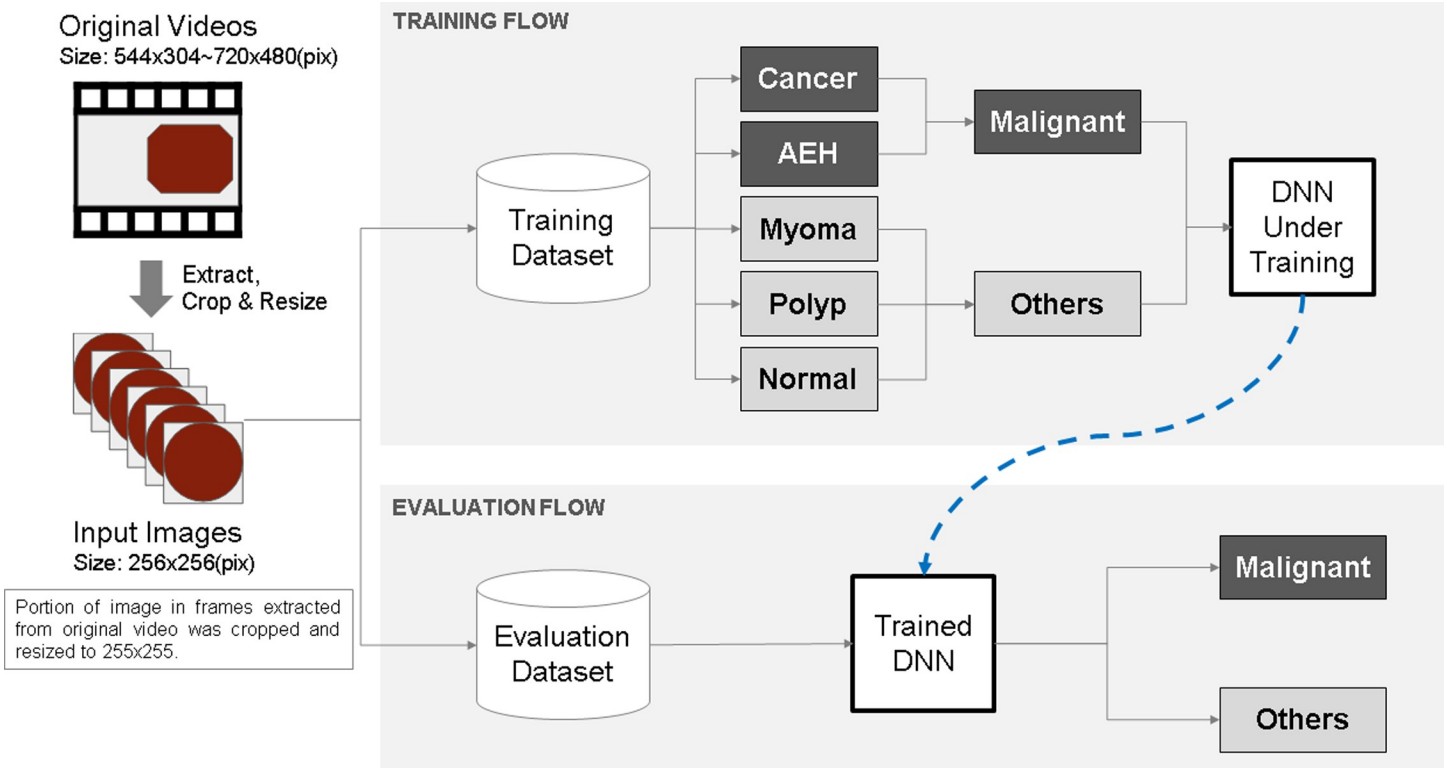

**Fig 2. Overall architecture of the model developed in this project.**

### Neural network types

As already stated, three different neural networks—Xception, MobileNetV2, and EfficientNetB0 —were adopted in this study to classify the images extracted from the video stream. These networks can exhibit relatively high accuracy with smaller size datasets and less expensive learning costs. We built these models using Keras implemented on TensorFlow and then trained them on an Intel core i7-9700 CPU + Nvidia GTX 1080ti GPU. The number of parameters used with each network is shown in S4 Table. The time spent to learn 3,000,000 images is shown in Fig 4B.

The network structure of Xception is shown in S5 Table. The most unique feature of Xception is that it divides the normal convolutional network layer into micro-networks called Inception modules as much as possible and replaces them with "Depthwise Separable Convolution." The "Depthwise Separable Convolution" network structure divides the normal convolutional network into two network segments, Depthwise Convolution and Pointwise Convolution [19]. The network structure of MobileNetV2 is shown in S6 Table. The most unique feature of MobileNet is the adoption of the network layers called "Inverted Residual" widely to almost every network layer to reduce the total number of parameters [20]. The network structure of EfficientNetB0 is shown in S7 Table. The most unique feature of EfficientNet is the introduction of compound coefficients based on how the depth, width, resolution, etc. of the network within a convolutional network affect the performance of the model [21].

### Model generation—execution of training

Owing to the nature of neural networks, even when the same type of neural network is trained using the same dataset, each model yields a different accuracy. Therefore, in this study, we

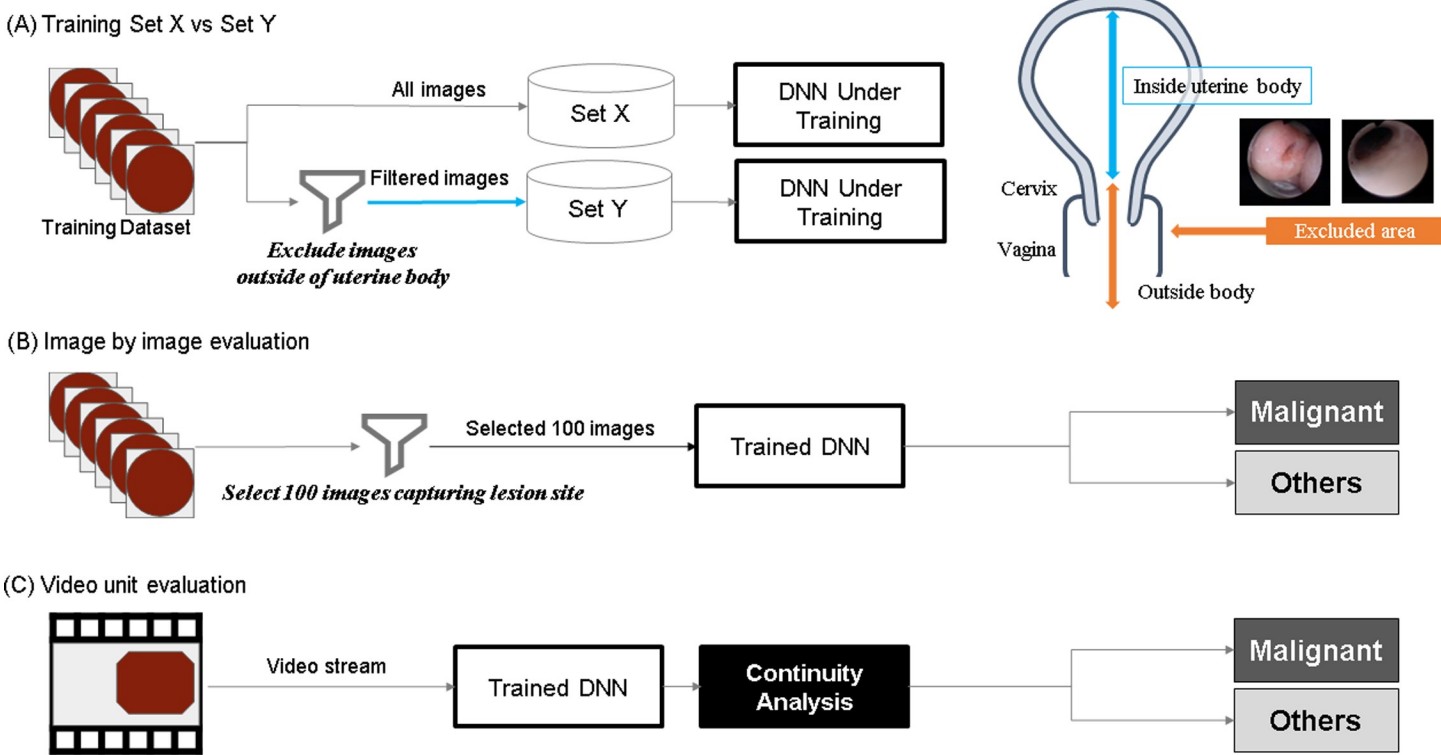

**Fig 3.** (A) Schematic of the training method: The training data pertaining to the malignant class were separated into two sets, Set X and Set Y. (B) Schematic of the evaluation method: image by image. (C) Schematic of the evaluation method: video unit. During image-by-image evaluation, 100 images that clearly included the lesion site were extracted from the hysteroscopic video of each patient diagnosed with a malignant tumor (Continuity analysis).

trained three types of DNN models six times using two datasets (Set X, Set Y), which were grouped into four training and evaluation pairs—A, B, C, and D. Thus, 144 ($3 \times 6 \times 2 \times 4$) trained models were acquired.

## Results

### Results of image by image evaluation

In this study, we first evaluated the accuracies of the predicted results obtained using each of the above-mentioned 144 models to each individual image. Subsequently, we calculated the average accuracy values by dividing the results into two groups based on the applicable data class and neural network type. Comparisons between the average prediction accuracies obtained for each dataset and network type are presented in Figs 4C and S1A and S8 Table. As can be realized, the difference between the average accuracy values (0.7891 and 0.8093, respectively) obtained for datasets X and Y equaled 0.0201, whereas that between the accuracy values obtained using the different network types equaled 0.0047 (0.7969 (minimum) and 0.8016 (maximum)) (S8 Table). As observed in this study, MobileNetV2 demonstrated the shortest learning time, whereas Xception required the longest learning duration—approximately thrice that required by MobileNetV2, as described in Fig 4B.

### Results of video-unit-based evaluation: Continuity analysis

As already stated, the continuity analysis method for use in hysteroscopy applications has been developed in this study to increase the diagnostic accuracy realized when performing video-

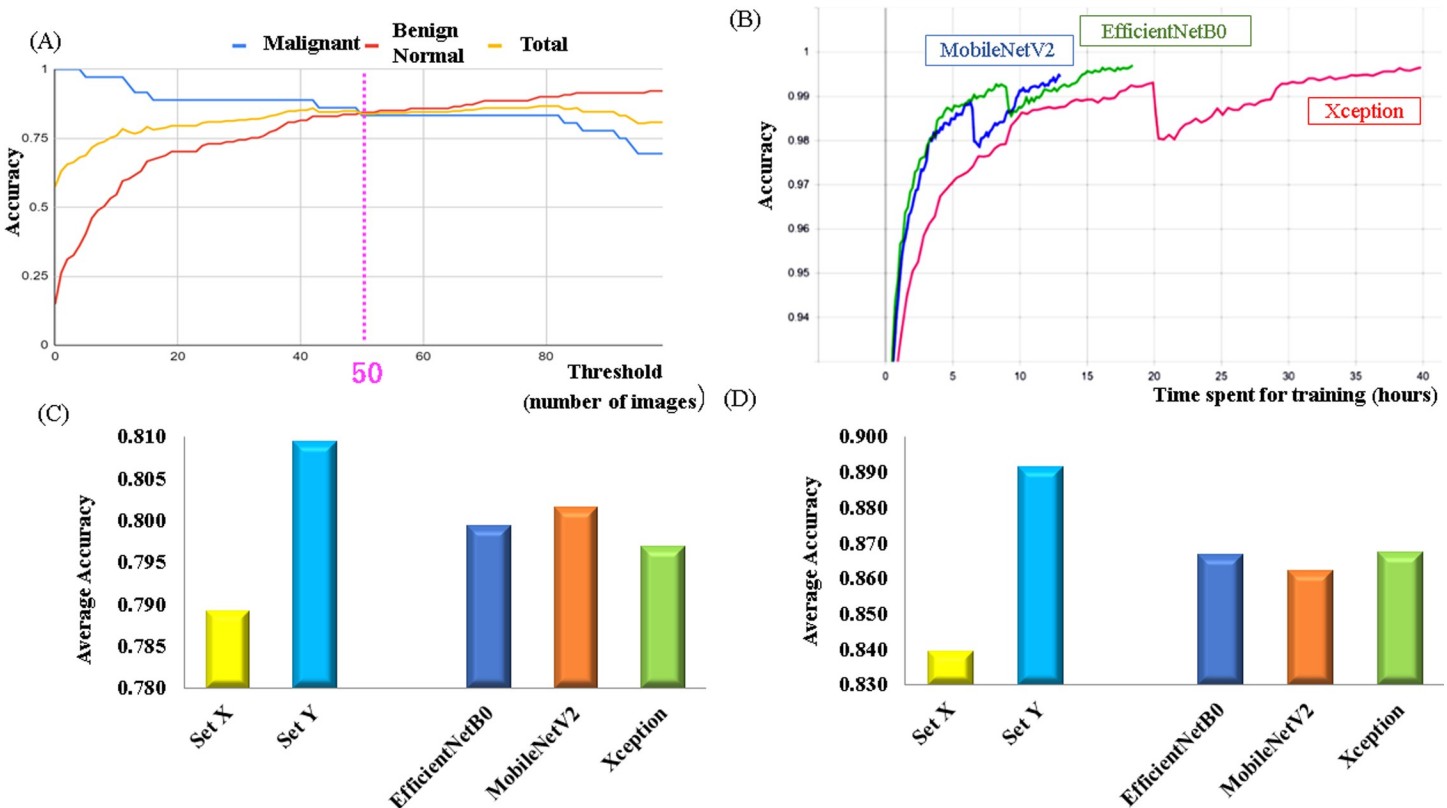

**Fig 4.** (A) Trend depicting accuracy displacement of malignant and benign diagnoses in accordance with threshold value for continuity analysis. (B) Comparison between learning times required by the three neural networks. The physical time depends on the computer specifications and image size; however, the ratio of the learning time required by each network is independent of such conditions.(C) Average accuracy values obtained via image-by-image-based predictions grouped in terms of dataset and network type. (D) Average accuracy values obtained via video-unit-based predictions grouped in terms of dataset and network type.

unit-based evaluations. As mentioned in the Materials and methods section, hysteroscopy video samples were considered representative of malignant tumors when 50 or more consecutive image frames extracted from them were classified as "Malignant." Comparisons between the average prediction accuracies obtained for each dataset and network type are presented in Figs 4D and S1B and S9 Table. As can be seen, the difference between the average accuracy values (0.8394 and 0.8913, respectively) obtained for datasets X and Y equaled 0.0512, whereas that between accuracy values obtained using the different network types equaled 0.0052 (0.8622 (minimum) and 0.8675 (maximum)) (Figs 4D and S1B and S9 Table).

## Evaluation of accuracy improvements realized by combining multiple models

Finally, we evaluated the improvement in diagnostic accuracy realizable by using a combination of multiple DNN models. The evaluation was performed using 72 models (6 iterations × 4 data pairs × 3 model types) trained using Set Y. The video-unit-based continuity-analysis method was used owing to its demonstrated superior performance compared to the image-by-image-based technique. The results of this evaluation (Fig 5 and Table 2) revealed that the combination of 72 models could classify cancers and AEH as part of the malignant group accuracies of 0.8571 and 1.000, respectively. Likewise, the diagnostic accuracies for myomas, endometrial polyps, and normal endometrium equaled 0.8571, 0.8500, and 0.9500, respectively.

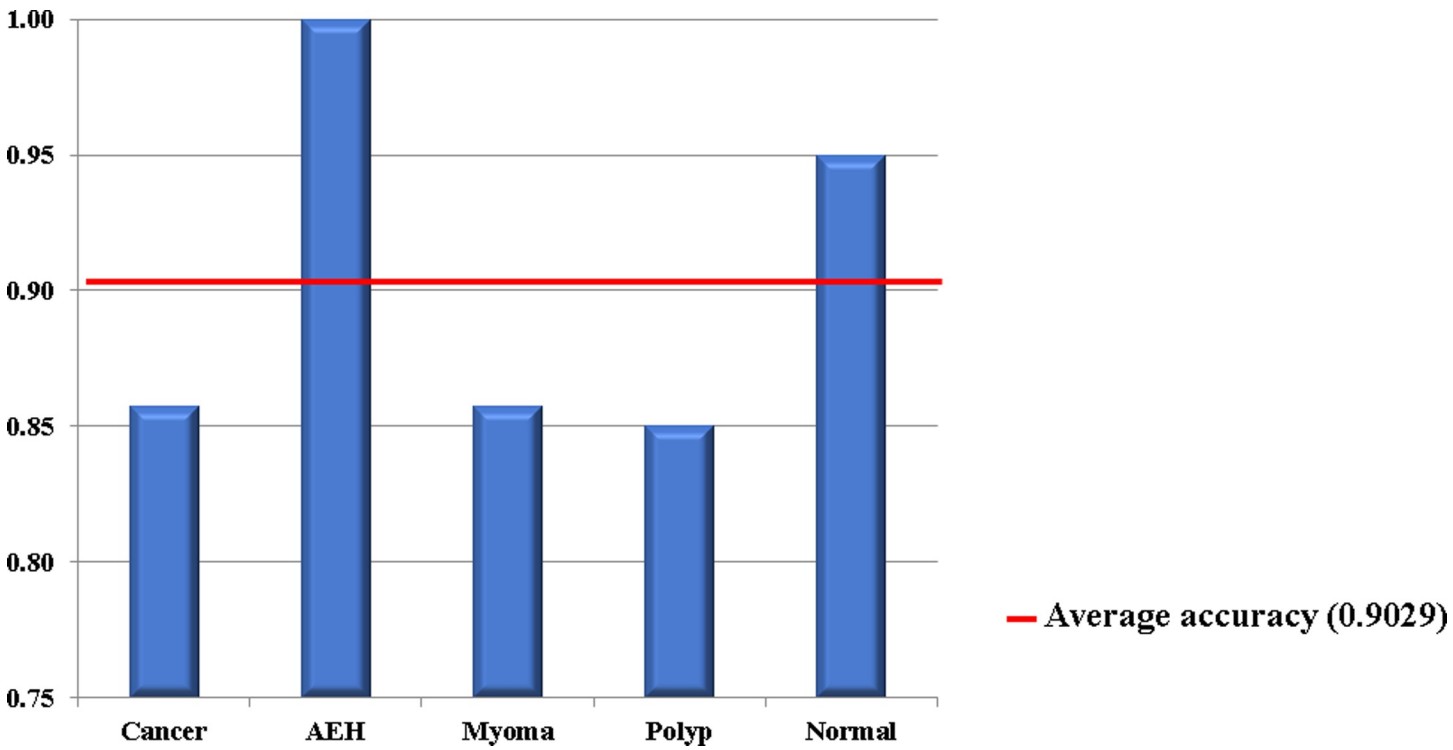

**Fig 5. Average diagnostic accuracies for different conditions obtained using combination of 72 trained deep neural network models.**

The overall average accuracy equaled 0.9029 with corresponding sensitivity and specificity values of 91.66% (95% confidence interval (CI) = 77.53–98.24%) and 89.36% (95% CI = 83.06–93.92%), respectively (Table 2). In addition, the value of F-score was 0.757. These results confirm the realization of superior diagnostic accuracy when using the combination of prediction models compared to their standalone utilization.

## Discussion

In this study, we aimed to develop a DNN-based automated system to detect the presence of endometrial tumors in hysteroscopic images. As observed in this study, an average diagnostic

**Table 2. Diagnosis results obtained using combination of 72 trained deep neural network models.**

| | Truth | Prediction | | Total | Correct | Sensitivity | Specificity | F-score | Accuracy | Average |
|---|---|---|---|---|---|---|---|---|---|---|
| | | Malignant | Others | | | | | | | |
| Cancer | Malignant | 18 | 3 | 21 | 18 | | | | 0.8571 | |
| AEH | Malignant | 15 | 0 | 15 | 15 | | | | 1 | |
| Myoma | Others | 3 | 18 | 21 | 18 | | | | 0.8571 | 0.9029 |
| Polyp | Others | 9 | 51 | 60 | 51 | | | | 0.85 | |
| Normal | Others | 3 | 57 | 60 | 57 | | | | 0.95 | |
| Total | | 48 | 129 | 177 | 159 | 0.9167 | 0.894 | 0.7857 | 0.8983 | |
| Correct | | 33 | 126 | | | | | | | |
| Precision | | 0.6875 | 0.9767 | | | | | | | |

AEH: Atypical endometrial hyperplasia.

accuracy exceeding 90% was realized when using the combination of 72 trained DNN models. Overall, we were able to realize a relatively high diagnostic accuracy, despite the consideration of only a limited number of endometrial cancer and AEH cases.

As described in the Introduction section, several deep-learning models for use in image-recognition applications have been developed in recent years. Additionally, their utilization in medical applications has been thoroughly investigated. For example, Esteva et al. [22] developed a deep-learning algorithm trained on a dataset comprising more than 129,000 images of over 2,000 different skin diseases. Subsequently, they evaluated whether their proposed classification system could successfully distinguish skin-cancer cases from those corresponding to benign skin diseases. They observed that their proposed system could demonstrate diagnostic performance on par with that proposed by a group of clinical specialists [22]. Automated systems that perform disease diagnoses by applying deep-learning models to endoscopic images, such as those captured by gastrointestinal endoscopes and cystoscopes, have been developed in recent years [17,18]. Although colorectal neoplastic polyps represent the precancerous lesions of colorectal cancer, their presence can be typically diagnosed by an endoscopist with the naked eye. However, the presence of these polyps can remain undetected in cases where they are either very small or possess shapes that make it difficult to identify them. Yamada et al. [18] developed a convolutional neural network-based deep-learning model that they applied to endoscopic images captured for approximately 5,000 cases; their proposed analysis yielded a polyps and precancerous-lesion detection rate of 98%.

In general, the application of deep-learning techniques to image-recognition problems requires collection of 100,000–1,000,000 images to constitute a viable training dataset. However, as described earlier, in the medical field it can be difficult to obtain such a large number of samples depending on diseases and circumstances. Because the diagnosis of cancer by hysteroscopy is not a common method, it is difficult to obtain a large number of samples from a single institution at present. Therefore, in recent AI research in the medical field, a major focus is to achieve a high accuracy rate with a small sample size; there are some reports that address this. For example, Sakai et al. [23] extracted small regions from a small number of endoscopic images obtained during the early stages of gastric cancer. Data expansion technology was utilized to increase the number of images to approximately 360,000. The application of a convolutional neural network to the said image dataset yielded positive and negative predictive values of 93.4% and 83.6%, respectively. A major limitation of this study is that the video stream contained a significant number of frames that did not capture the lesions to be identified [23]. Therefore, we deleted all frames that did not capture lesions in the extracted image in Set Y. However, even frames that do not depict lesions might include malignant-tumor-specific features, such as cloudy uterine luminal fluid. Moreover, even when the degree of cloudiness is too small to be recognized by the naked eye, it can be accurately recognized by computers. Therefore, we divided the learning data into two datasets—Set X and Set Y. As described in the Results section, the results obtained using Set Y yielded a higher diagnostic accuracy compared to Set X. This suggests that the diagnostic accuracy can be improved by exclusively analyzing the lesion sites instead of all extracted images comprising the dataset. Moreover, given the limited use of hysteroscopy in medical practice and the need for consideration of several training cases to leverage the existing deep-learning models for analysis of medical images, we developed a continuity-analysis method based on a combination of neural networks. The proposed method demonstrates the realization of high diagnostic accuracies, despite the use of a limited training dataset.

It is noteworthy that accuracies of 90% or more can be obtained with such a small sample size. The proposed system is our original idea and is the most significant aspect of this research. The method can also be applied to other types of medical images with fewer samples,

as well as hysteroscopic images. While gastrointestinal endoscopy is commonly used in the diagnosis of gastric and colorectal cancers, in general, hysteroscopy is seldom used in the diagnosis of endometrial cancer. However, our previous study [8] demonstrates the usefulness of hysteroscopy in the diagnosis of endometrial cancer. Therefore, if a hysteroscopy-based automated system employing deep-learning models is established for clinical diagnosis of endometrial cancer, an increase in the use of hysteroscopes, can be expected as well.

As already mentioned, early diagnosis of endometrial cancer can help patients retain their fertility, and it may even eliminate the need for post-therapy, which involves the use of anti-cancer drugs and radiation therapy, despite a surgery being performed [1,3,24]. The diagnostic system presented in this paper demonstrates the potential to be an effective system for accurate diagnosis of endometrial cancer in future. In the future, a large-scale study will be conducted using the algorithm established in this study. Therefore, the current study is a pilot to determine whether large-scale research is possible. Notably, implementation of the proposed system in its entirety is necessary to improve the positive and negative predictive values to around 100%. To facilitate high-accuracy diagnosis, it is necessary to (1) use a large number of images as well as add notations to all existing and new images and (2) develop a high-accuracy engine. Another limitation of this study is that although the use of the combinational model facilitated realization of a high diagnostic accuracy, the capacity was large when considering medical device development. Thus, the development of a more compact system must be pursued to accommodate a large number of cases. However, as mentioned before, it is difficult to significantly increase the number of hysteroscopic images in a single facility, and as future study, we aim to increase the number of samples by using this system in a multi-facility joint research collaboration.

To the best of our knowledge, this study represents the first attempt toward the diagnosis of endometrial cancer using a combination of deep learning and hysteroscopy. Although two studies [25,26] concerning hysteroscopy and deep learning have been previously reported, they exclusively concern uterine myomas and *in vitro* fertilization, respectively, and they have not yet been used in endometrial-cancer diagnosis.

As described in the Materials and methods section, three neural networks—Xception [19], MobileNetV2 [20], and EfficientNetB0 [21]—were used in this study to classify frame images extracted from video samples. These networks were selected because they are computationally inexpensive and demonstrate high accuracy, thereby facilitating real-time diagnosis while incurring low manufacturing costs. Therefore, it is important to clarify the relationship between the execution speed and neural network accuracy. From the viewpoint of the future development of deep-learning-based medical devices, it is necessary to compare real-time and post-hysteroscopy analyses. Additionally, we examined the images for which the deep-learning algorithms considered in this study could not perform an accurate diagnosis. The following two features were identified—(1) the flatness of the tumor and (2) difficulty in tumor identification due to excessive bleeding. The issues can be resolved by increasing the number of images in the training dataset. However, the size of the tumor cannot be considered a cause of error. In the future, when considering a large number of cases, it is necessary to perform subgroup analysis in accordance with the patient's age, stage of the disease, histology, etc. Moreover, it is necessary to make a comparison with hysteroscopic specialists.

## Conclusion

The challenge in medical AI research is to develop a system analysis method for improving the accuracy with a small number of samples. It is noteworthy that a high accuracy for diagnosis of endometrial cancer can be obtained with such a small sample in this study and we believe that

the capability of the basic system has been established in this study. The accuracy rate of conventional diagnostic techniques, such as pathological diagnoses by curettage and cytology, is low, and screening for endometrial cancer has not been established. In the future, multi-institutional joint research should be conducted to develop this system. If this system is properly developed, it can be utilized for the screening of endometrial cancer.

## Supporting information

**S1 Fig.** (A) Diagnostic accuracy realized when applying the neural networks on individual datasets. Image-classification accuracy was compared using dataset–neural-network combination. (B) Diagnostic accuracy realized when employing proposed continuity analysis using dataset–neural-network combination.
(TIF)

**S1 Table. Stages and histological types endometrial cancer identified in patients recruited in this study.**
(DOCX)

**S2 Table. Training and evaluation data in this study.**
(DOCX)

**S3 Table. Datasets used in this study.**
(DOCX)

**S4 Table. Number of parameters of each network.**
(DOCX)

**S5 Table. Network structure of EfficientNet B0.**
(DOCX)

**S6 Table. Network structure of MobileNet V2.**
(DOCX)

**S7 Table. Network structure of Xception.**
(DOCX)

**S8 Table. Average accuracies obtained through image-by-image-based predictions grouped in terms of dataset and network types.**
(DOCX)

**S9 Table. Average accuracies obtained through video-unit-based predictions grouped in terms of dataset and network types.**
(DOCX)

## Acknowledgments

The authors thank Editage for English language editing (https://www.editage.com/).

## Author Contributions

**Conceptualization:** Yu Takahashi, Kenbun Sone, Katsuhiko Noda, Kaname Yoshida, Katsutoshi Oda, Yutaka Osuga, Tomoyuki Fujii.

**Data curation:** Yu Takahashi, Yusuke Toyohara, Kosuke Kato, Futaba Inoue, Haruka Nishida, Yuichiro Miyamoto, Tetsushi Tsuruga, Osamu Hiraike-Wada, Masanori Maruyama.

**Formal analysis:** Yu Takahashi, Katsuhiko Noda, Yusuke Toyohara, Asako Kukita, Ayumi Taguchi.

**Investigation:** Yu Takahashi, Kenbun Sone, Katsuhiko Noda, Kaname Yoshida.

**Methodology:** Yu Takahashi, Kenbun Sone, Kaname Yoshida, Yusuke Toyohara, Kosuke Kato.

**Project administration:** Katsuhiko Noda.

**Resources:** Michihiro Tanikawa, Kazunori Nagasaka, Yasushi Hirota.

**Supervision:** Takayuki Iriyama, Katsutoshi Oda, Yutaka Osuga, Tomoyuki Fujii.

**Validation:** Yu Takahashi, Kenbun Sone, Katsuhiko Noda, Kaname Yoshida, Yusuke Toyohara, Kazunori Nagasaka, Yoko Matsumoto, Masanori Maruyama.

**Visualization:** Yusuke Toyohara.

**Writing – original draft:** Yu Takahashi, Kenbun Sone, Katsuhiko Noda, Kaname Yoshida, Yusuke Toyohara, Asako Kukita, Ayumi Taguchi, Kazunori Nagasaka, Yutaka Osuga, Tomoyuki Fujii.

**Writing – review & editing:** Kosuke Kato, Futaba Inoue, Haruka Nishida, Yuichiro Miyamoto, Michihiro Tanikawa, Tetsushi Tsuruga, Takayuki Iriyama, Yoko Matsumoto, Osamu Hiraike-Wada, Katsutoshi Oda, Masanori Maruyama.

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
