## [Decision Letter · Decision Letter 0]

7 Dec 2020

PONE-D-20-34280

Automated system for diagnosing endometrial cancer by adopting deep-learning technology in hysteroscopy

PLOS ONE

Dear Dr. Sone,

Thank you for submitting your manuscript to PLOS ONE. After careful consideration, we feel that it has merit but does not fully meet PLOS ONE’s publication criteria as it currently stands. Therefore, we invite you to submit a revised version of the manuscript that addresses the points raised during the review process.

We look forward to receiving your revised manuscript.

Kind regards,

Tao Song

Academic Editor

PLOS ONE

Journal Requirements:

"Kenbun Sone has a joint research agreement with Predicthy LLC. The other authors have no competing interests to disclose"

We note that one or more of the authors are employed by a commercial company: Predicthy LLC.

2.1. Please provide an amended Funding Statement declaring this commercial affiliation, as well as a statement regarding the Role of Funders in your study. If the funding organization did not play a role in the study design, data collection and analysis, decision to publish, or preparation of the manuscript and only provided financial support in the form of authors' salaries and/or research materials, please review your statements relating to the author contributions, and ensure you have specifically and accurately indicated the role(s) that these authors had in your study. You can update author roles in the Author Contributions section of the online submission form.

2.2. Please also provide an updated Competing Interests Statement declaring this commercial affiliation along with any other relevant declarations relating to employment, consultancy, patents, products in development, or marketed products, etc.  

Reviewers' comments:

Reviewer's Responses to Questions

**Comments to the Author**

1. Is the manuscript technically sound, and do the data support the conclusions?

Reviewer #1: Yes

Reviewer #2: Partly

2. Has the statistical analysis been performed appropriately and rigorously? 

Reviewer #1: Yes

Reviewer #2: Yes

3. Have the authors made all data underlying the findings in their manuscript fully available?

Reviewer #1: Yes

Reviewer #2: Yes

4. Is the manuscript presented in an intelligible fashion and written in standard English?

Reviewer #1: Yes

Reviewer #2: Yes

5. Review Comments to the Author

Reviewer #1: This work aims to establish deep learning models for classifying the presence of endometrial tumors in hysteroscopic images. And an average diagnostic accuracy exceeding 90% was realized when using the combination of 72 trained DNN models. However, I have the following concerns:

1)I am a bit curious why they use this deep learning architecture for endometrial tumors detection, rather than shallow machine learning models.

2)There are several errors in this manuscript, such as “The corresponding sensitivity and specificity equaled 91.66% and 89.36, respectively”. Is it 89.36%? The authors should double check the manuscript.

3)The manuscript should give the overall model architecture.

4)The metric method for the model is too simple, the author should add more metric method. Please refer to several literatures, such as:

Pang Shanchen, Ding Tong, Qiao Sibo, Meng Fan, Wang Shuo, Li pibao, WangXun . A novel YOLOv3-arch model for identifying cholelithiasis and classifying gallstones on CT images,2019, Plos one, 6(14):e0217647.DOI: 10.1371

Wang Shudong, Dong Liyuan, Wang Xun, Wang Xingguang. Classification of Pathological Types of Lung Cancer from CT Images by Deep Residual Neural Networks with Transfer Learning Strategy. Open Medicine, 2020, 15(1): 190-197.

Shanchen Pang, Yaqin Zhang, Mao Ding, Xun Wang, Xianjin Xie. A Deep Model for Lung Cancer Type Identification by Densely Connected Convolutional Networks and Adaptive Boosting. IEEE Access 2020,8: 4799-4805.

Shanchen Pang, Fan Meng, Xun Wang, et al. VGG16-T: A Novel Deep Convolutional Neural Network with Boosting to Identify Pathological Type of Lung Cancer in Early Stage by CT Images, International Journal of Computational Intelligence Systems. Vol.13(1), pp. 771-780, 2020.

Reviewer #2: In the paper, authors present an artificial-intelligence-based system to detect the regions affected by endometrial cancer automatically from hysteroscopic images. The diagnosis accuracy is increased. However, there are some details that can be improved.

The models used in the paper are not presented well.

The set of threshold value is 50, maybe you can explain some details about that.

The writing of the paper should be taken care. For example, the text size on the tables, the text-transform on subtitle of page 7.

6. PLOS authors have the option to publish the peer review history of their article (what does this mean?). If published, this will include your full peer review and any attached files.

Reviewer #1: No

Reviewer #2: No

---

## [Author Response · Author response to Decision Letter 0]

12 Jan 2021

Reviewer #1:

This work aims to establish deep learning models for classifying the presence of endometrial tumors in hysteroscopic images. And an average diagnostic accuracy exceeding 90% was realized when using the combination of 72 trained DNN models. However, I have the following concerns:

Comment1

I am a bit curious why they use this deep learning architecture for endometrial tumors detection, rather than shallow machine learning models.

Response 1

We appreciate your critical comments and useful suggestions. Deep learning is highly anticipated in the medical field because deep learning techniques are particularly suitable for image analysis. They can be used for classification, image quality improvement, and segmentation of medical images. In contrast, shallow machine learning is not suitable for image recognition. We have added this information to the revised manuscript considering your comment (Lines 66-69).

Comment2

There are several errors in this manuscript, such as “The corresponding sensitivity and specificity equaled 91.66% and 89.36, respectively”. Is it 89.36%? The authors should double check the manuscript.

Response 2 

We appreciate your critical comments and useful suggestions. It is 89.36%（Lines36）. We have corrected the oversight.

Comment3

The manuscript should give the overall model architecture.

Response 3

We appreciate your critical comments and useful suggestions. We have added the overall architecture of the model (Figure2) in accordance with your suggestion.

Comment4

The metric method for the model is too simple, the author should add more metric method. Please refer to several literatures, such as:

Pang Shanchen, Ding Tong, Qiao Sibo, Meng Fan, Wang Shuo, Li pibao, WangXun . A novel YOLOv3-arch model for identifying cholelithiasis and classifying gallstones on CT images,2019, Plos one, 6(14):e0217647.DOI: 10.1371

Wang Shudong, Dong Liyuan, Wang Xun, Wang Xingguang. Classification of Pathological Types of Lung Cancer from CT Images by Deep Residual Neural Networks with Transfer Learning Strategy. Open Medicine, 2020, 15(1): 190-197.

Shanchen Pang, Yaqin Zhang, Mao Ding, Xun Wang, Xianjin Xie. A Deep Model for Lung Cancer Type Identification by Densely Connected Convolutional Networks and Adaptive Boosting. IEEE Access 2020,8: 4799-4805.

Shanchen Pang, Fan Meng, Xun Wang, et al. VGG16-T: A Novel Deep Convolutional Neural Network with Boosting to Identify Pathological Type of Lung Cancer in Early Stage by CT Images, International Journal of Computational Intelligence Systems. Vol.13(1), pp. 771-780, 2020.

Response 4 

We appreciate your critical comments and useful suggestions. We have added the metric methods in accordance with your comments. F-score and Precision have been added to Table 2. In addition, the description and structure of each network are also given (Tables S4, S5, S6, S7, Lines 164-179).

Reviewer #2:

In the paper, authors present an artificial-intelligence-based system to detect the regions affected by endometrial cancer automatically from hysteroscopic images. The diagnosis accuracy is increased. However, there are some details that can be improved.

Comment1

The models used in the paper are not presented well.

Response 1 

We appreciate your critical comments and useful suggestions. We have added the overall architecture of the model (Fgure2) to provide further details of the model used. In addition, the description and structure of each network are also given (Tables S4, S5, S6, S7, Lines 164-179).

Comment2

The set of threshold value is 50, maybe you can explain some details about that.

Response 2 

We appreciate your critical comments and useful suggestions. The threshold was taken from the points where the malignant score intersects with the other scores rather than the point where the average of two scores was the best, because the threshold should be set lower to reduce oversight cases in the actual clinical devices. We have added this information to the revised manuscript considering your comment (Lines 152-154).

Comment3

The writing of the paper should be taken care. For example, the text size on the tables, the text-transform on subtitle of page 7.

Response 3 

We appreciate your critical comments and useful suggestions. We have revised the manuscript in accordance with your suggestion and PLOS ONE's style requirements.

---

## [Decision Letter · Decision Letter 1]

1 Mar 2021

Automated system for diagnosing endometrial cancer by adopting deep-learning technology in hysteroscopy

PONE-D-20-34280R1

Dear Dr. Sone,

We’re pleased to inform you that your manuscript has been judged scientifically suitable for publication and will be formally accepted for publication once it meets all outstanding technical requirements.

Kind regards,

Tao Song

Academic Editor

PLOS ONE

Additional Editor Comments (optional):

Reviewers' comments:

Reviewer's Responses to Questions

**Comments to the Author**

1. If the authors have adequately addressed your comments raised in a previous round of review and you feel that this manuscript is now acceptable for publication, you may indicate that here to bypass the “Comments to the Author” section, enter your conflict of interest statement in the “Confidential to Editor” section, and submit your "Accept" recommendation.

Reviewer #1: All comments have been addressed

Reviewer #2: All comments have been addressed

2. Is the manuscript technically sound, and do the data support the conclusions?

Reviewer #1: Yes

Reviewer #2: Yes

3. Has the statistical analysis been performed appropriately and rigorously? 

Reviewer #1: Yes

Reviewer #2: Yes

4. Have the authors made all data underlying the findings in their manuscript fully available?

Reviewer #1: Yes

Reviewer #2: Yes

5. Is the manuscript presented in an intelligible fashion and written in standard English?

Reviewer #1: Yes

Reviewer #2: Yes

6. Review Comments to the Author

Reviewer #1: Thanks for your efforts, all comments have been addressed by the authors, so, I recommand to accpet the manuscript.

Reviewer #2: In the paper, authors present an artificial-intelligence-based system to detect the regions affected by endometrial cancer automatically from hysteroscopic images. The diagnosis accuracy is increased. The authors replied well to the suggestions I proposed. It can be accepted.

7. PLOS authors have the option to publish the peer review history of their article (what does this mean?). If published, this will include your full peer review and any attached files.

Reviewer #1: No

Reviewer #2: No

---

## [Editor Report · Acceptance letter]

5 Mar 2021

PONE-D-20-34280R1 

Automated system for diagnosing endometrial cancer by adopting deep-learning technology in hysteroscopy 

Dear Dr. Sone:

I'm pleased to inform you that your manuscript has been deemed suitable for publication in PLOS ONE. Congratulations! Your manuscript is now with our production department. 

Kind regards, 

on behalf of

Dr. Tao Song 

Academic Editor

PLOS ONE